# A genome-scale screen reveals context-dependent ovarian cancer sensitivity to miRNA overexpression

Benjamin B Shields[1], Chad V Pecot[2], Hua Gao[1], Elizabeth McMillan[1], Malia Potts[1], Christa Nagel[3], Scott Purinton[3], Ying Wang[4], Cristina Ivan[4], Hyun Seok Kim[5], Robert J Borkowski[1], Shaheen Khan[6], Cristian Rodriguez-Aguayo[2], Gabriel Lopez-Berestein[2], Jayanthi Lea[3], Adi Gazdar[7], Keith A Baggerly[4], Anil K Sood[2] & Michael A White[1],*

## Abstract

Large-scale molecular annotation of epithelial ovarian cancer (EOC) indicates remarkable heterogeneity in the etiology of that disease. This diversity presents a significant obstacle against intervention target discovery. However, inactivation of miRNA biogenesis is commonly associated with advanced disease. Thus, restoration of miRNA activity may represent a common vulnerability among diverse EOC oncogenotypes. To test this, we employed genome-scale, gain-of-function, miRNA mimic toxicity screens in a large, diverse spectrum of EOC cell lines. We found that all cell lines responded to at least some miRNA mimics, but that the nature of the miRNA mimics provoking a response was highly selective within the panel. These selective toxicity profiles were leveraged to define modes of action and molecular response indicators for miRNA mimics with tumor-suppressive characteristics *in vivo*. A mechanistic principle emerging from this analysis was sensitivity of EOC to miRNA-mediated release of cell fate specification programs, loss of which may be a prerequisite for development of this disease.

**Keywords**  cancer; cancer genetics; microRNA; miRNA; ovarian cancer
**Subject Categories**  Chromatin, Epigenetics, Genomics & Functional Genomics; Cancer
**Mol Syst Biol. (2015) 11: 842**

## Introduction

Epithelial ovarian cancer (EOC) is the most lethal gynecologic malignancy in the United States (Siegel *et al*, 2012). Recent advances in treatment of this disease have been limited to empirical optimization of chemotherapeutic agents and improved delivery of drugs (Armstrong *et al*, 2006). While these have yielded measurable improvements in overall survival of ovarian cancer patients, there is an urgent need for novel treatment modalities. A greater understanding of the linchpin biology of this disease would likely help provide inroads toward the development of new therapies.

Multiple public and private efforts have focused on large-scale annotation of the landscape of genomic alterations associated with EOC. These studies have detected over 60 tumor-acquired mutations per patient. Though mutation of the p53 tumor suppressor has been identified as an almost universal characteristic (95% of ovarian tumors), all other somatic mutations have been found to occur in 3–6% of tumors or less (Kan *et al*, 2010; Cancer Genome Atlas Research, 2011). In combination with this diversity of somatic nucleotide variation, pervasive and recurrent copy number variation has been detected (Etemadmoghadam *et al*, 2009; Cancer Genome Atlas Research N, 2011), giving rise to the notion that ovarian tumor progression is driven by a "turbulent genome".

The seemingly enormous diversity of molecular etiology of EOC poses a significant challenge to intervention target discovery and is fueling efforts to identify common biological vulnerabilities that occupy the nexus of diverse EOC genomes. A compelling candidate is defective miRNA biogenesis and function. When measured by quantitative PCR, Dicer and Drosha, the RNases required for miRNA processing, show decreased expression in over half of ovarian tumors sampled, and high Dicer expression correlates with remarkable patient survival (Merritt *et al*, 2008). However, we note that microarray-based tools have failed to uncover this relationship in larger cohorts (Gyorffy *et al*, 2013; Madden *et al*, 2014). Dicer is a haploinsufficient tumor suppressor in mice, and compound deletion of Dicer and the PTEN tumor suppressor is sufficient to induce spontaneous epithelial ovarian cancer (Kumar *et al*, 2009; Kim *et al*,

---

1  Departments of Cell Biology, University of Texas Southwestern Medical Center, Dallas, TX, USA
2  Center for RNA interference and Non-Coding RNA, MD Anderson Cancer Center, Houston, TX, USA
3  Obstetrics and Gynecology, University of Texas Southwestern Medical Center, Dallas, TX, USA
4  Department of Bioinformatics and Computational Biology, MD Anderson Cancer Center, Houston, TX, USA
5  Severance Biomedical Science Institute, Yonsei University College of Medicine, Seoul, Korea
6  Immunology, University of Texas Southwestern Medical Center, Dallas, TX, USA
7  Pathology, University of Texas Southwestern Medical Center, Dallas, TX, USA
*Corresponding author. Tel: +1 214 648 4212; Fax: +1 214 648 5814; E-mail: michael.white@utsouthwestern.edu

2012). Together, these observations indicate miRNA production may be generally deleterious to ovarian tumor initiation and/or progression, perhaps through translational suppression of tumor promoting gene products. Of note, the 3′-untranslated regions (3′-UTRs) of many mRNAs are clipped in some cancer cell types, which can release oncogenes from miRNA regulation (Mayr & Bartel, 2009). Thus, sensitivity to restoration of miRNA activity may represent a common vulnerability among diverse EOC oncogenotypes.

To test the commonality of sensitivity to miRNA activity in EOC, we examined the consequence of introducing each of 400 miRNA mimics on the viability of each of 16 ovarian cancer cell lines, telomerase-immortalized ovarian surface epithelial cells, and hepatocytes. Selectively toxic mimics were recovered across the panel, the majority of which displayed highly individualized activity. Clinical correlations and mechanistic follow-up in multiple disease lineages indicated that the idiosyncratic miRNA mimic toxicity profiles were a consequence of fractional representation of biologically relevant ovarian cancer subtype/miRNA relationships that are more commonly encountered in other disease sites. Rare mimics targeting common vulnerabilities in the EOC panel corresponded to miR-517a and miR-124. miR-517a toxicity was primarily accounted for by its target ARCN1, a component of the COPI complex, and effectively impaired xenograft tumor growth when administered *in vivo*. miR-124 toxicity was primarily accounted for by its target SIX4, a homeobox transcription factor, which was also validated *in vivo*. A convergent mechanistic principle derived from this analysis was the common vulnerability of EOC to miRNA-mediated release of aberrant cell differentiation programs, loss of which may be a prerequisite for development of disease.

## Results

### A genomewide screen for miRNAs with antineoplastic potential in ovarian cancer

For broad-scale interrogation of the selective consequences of gain-of-function microRNA activity, in ovarian cancer cell regulatory contexts, we combined a genome-scale synthetic miRNA collection with a panel of ovarian cancer cell lines representative of the genomic diversity found in this disease. The miRNA mimic collection corresponded to 400 unique human miRNA annotated in miRBase8-10 (Dataset EV1). As a test-bed within which to assess selective inhibition of cancer cell viability, we collected a panel of 16 ovarian tumor-derived cell lines together with non-tumorigenic telomerase-immortalized ovarian surface epithelial cells and spontaneously immortalized hepatocytes. This panel included commonly employed laboratory lines, a matched pair of chemoresponsive and chemoresistant lines from the same patient (PEO1, PEO4), and newly derived low-passage non-clonal cultures isolated from the malignant peritoneal effusions of 3 patients with high-grade serous papillary adenocarcinoma of the ovary (HCC5012, HCC5019, HCC5030, Table EV1). Each miRNA mimic was introduced into each cell line using optimized transient transfection protocols (Fig EV1A, Table EV1), and consequent effects on cell viability were measured 120 h later from biological triplicates. Standard deviation distributions indicated high reproducibility among biological triplicates across the cell line panel (Fig EV1B, black curve), and high

phenotypic correlation among miRNA seed family members (Fig EV1B, red curve) relative to the total phenotypic variation (Fig EV1B, blue curve). Mean viability scores were normalized against position and batch effects and converted to $z$-scores to facilitate inter-line comparisons (Ho *et al*, 2012; Ward *et al*, 2012; Singh *et al*, 2013) (Dataset EV1). Affinity propagation clustering (APC) (Frey & Dueck, 2007; Witkiewicz *et al*, 2015) was used to delineate deterministic patterns of commonality among the miRNA mimic phenotypes across the cell line panel (Fig EV1C and Dataset EV11) and among the cell line responses to the miRNA mimic library (Fig EV1D). At least 50 phenotypic miRNA clusters were recovered which corresponded to five distinct cell line clusters (Fig EV1). APC of available whole-genome transcript profiles (Barretina *et al*, 2012) suggested at least 4 expression subtypes are present within the cell panel (Fig EV1E). However, these clusters had unimpressive correspondence to miRNA viability phenotype-based clusters (Fig EV1F) indicating global gene expression phenotypes, considered as a whole, did not specify selective response to the miRNA mimic library.

A total of 108 miRNA mimics, corresponding to 94 unique mature microRNA sequences, reduced cell viability two standard deviations below the mean ($z$-score $\leq -2$) in at least one cell line screened (Dataset EV1). Activity profiles, as visualized by two-way unsupervised hierarchical clustering, indicated a wide variation of selectivity patterns and potencies (Fig 1A). Of note, the most common miRNA mimic phenotype was idiosyncratic activity within the panel. About 80% of the miRNA mimics recovered in the screen significantly reduced the viability of only 1 or 2 cell lines, and no mimic reduced viability in more than 9 cell lines (Fig 1B). We considered the possibility that the selective activity profiles may be a consequence of fractional representation of biologically relevant ovarian cancer subtype/miRNA relationships within the test-bed, a consequence of artificial diversity from clonal genetic divergence *in vitro*, or a combination of the two. To help evaluate this, we first queried patient outcome data for the presence of significant clinical correlations to miRNAs with selective activity against the most resistant (and therefore least prone to noise from multiplicity of testing) cell line screened, SKOV3 (Fig EV2A). The expression of miRNAs corresponding to the top 5% of miRNA mimics with selective toxicity in SKOV3 (Dataset EV1) was evaluated in tumors from two independent ovarian cancer patient cohorts. We found that patients with higher expression of miR-146a and miR-505 have significantly increased overall survival with median overall survival times of 17.1 months and 10.4 months, respectively (Figs 1C and EV2D and E). The selective activity of miR-146a and miR-505 in SKOV3 was independent of transfection efficiencies or endogenous miR expression (Fig EV2B and C). Notably, recent studies indicate that both miR-146a and miR-505 have antitumorigenic activities in cell models of breast and lung cancer (Verduci *et al*, 2010; Yamamoto *et al*, 2011; Chen *et al*, 2013a). Next, we focused on miRNA mimics that selectively inhibited viability of the non-clonal short-term ascites-derived cultures (HCC5030, HCC5012, HCC5019), which should be least prone to *in vitro* genetic drift. We found significant enrichment ($P < 0.05$ by hypergeometric density distribution) of the miRNAs corresponding to these mimics among those miRNAs demonstrated to be downregulated in human serous ovarian tumors (Iorio *et al*, 2007) (Fig 1D). The clinical correlations of idiosyncratic hits, with

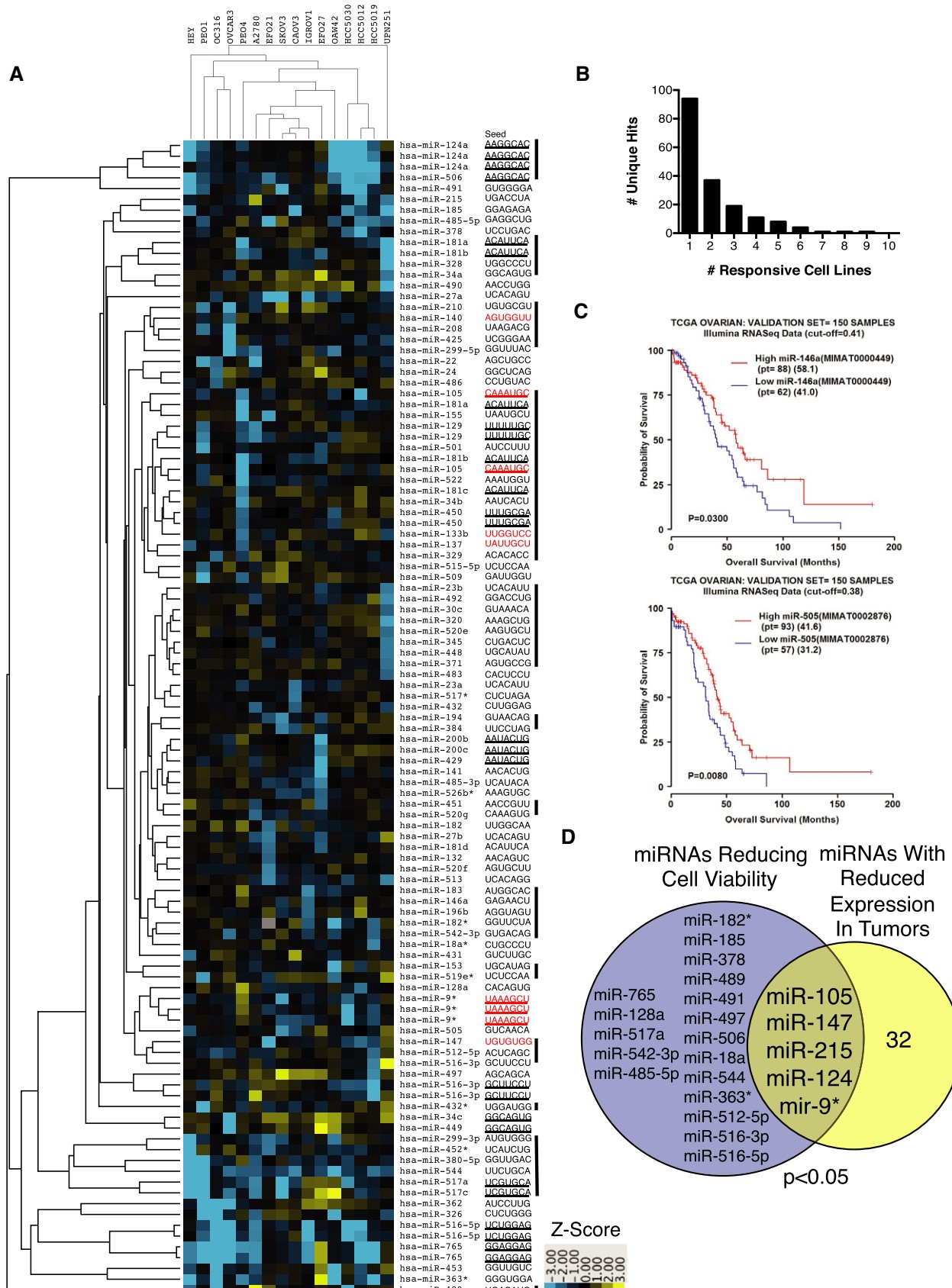

**Figure 1.**

**Figure 1.   Public and private miRNA vulnerabilities among ovarian cancer cell lines.**

A   Two-way unsupervised hierarchical cluster of *z*-score distributions by Euclidian distance. Any miRNA mimic with a *z*-score ≤ −2 in at least one cell line was included. Seed sequences of each miRNA mimic are on the right. Bars correspond to arbitrary cluster boundaries. Seeds in red correspond to miRNAs with decreased expression in serous, clear cell, or endometrioid ovarian cancer relative to normal tissue (Iorio *et al*, 2007).

B   The histogram indicates the number of non-redundant hits binned according to the number of responsive cell lines.

C   Expression of miRNAs miR-146a and miR-505 correlated with overall survival in ovarian cancer patients. The validation cohort (*n* = 150 samples) is shown. See Fig EV1 for the training cohort.

D   Intersection of miRNA sensitivities in the non-clonal short-term cultures and miRNAs with reduced expression in serous ovarian cancer (Iorio *et al*, 2007). *P*-value from hypergeometric distribution.

patient prognosis or molecular features from patient samples, suggest they can reflect germane ovarian tumor biology.

## miR-155 and miR-181b sensitivity is specified by intolerance to epithelial/mesenchymal transition

To begin to define mechanisms underpinning selective sensitivity to miRNA mimic exposure, we first focused on the patient-matched PEO1 and PEO4 cell lines. Derived later in the patient's treatment, the PEO4 cell line is a model for recurrent, platinum-resistant EOC (Fig EV3A). Cytogenetic analyses indicate these cell lines descended from a common ancestor as opposed to arising through direct linear descent (Wolf *et al*, 1987; Cooke *et al*, 2010; Stronach *et al*, 2011). Thus, these two cell lines provide a unique opportunity to investigate acquired vulnerabilities within a model of a single patient's recurrent disease. A scatter plot of the *z*-scores of each mimic from the PEO1 and PEO4 viability screens revealed two remarkably distinct tails of activity predominantly corresponding to miRNA-induced inhibition of viability in only one line or the other (Fig EV3B).

To examine whether differential miRNA mimic responsiveness corresponded to differential expression of the corresponding endogenous miRNAs, global miRNA expression in PEO1 and PEO4 was quantitated by Illumina array relative to that observed in non-tumorigenic human ovarian surface epithelium (HOSE) cells (GEO Reference GSE67329). miRNA toxicity was not solely a function of its presence or absence in either cell line (Fig EV3C and D). Although numerous significant differences in miRNA expression between PEO1 and PEO4 were identified (Datasets EV2 and EV8), there was no detectable correlation with selective miRNA mimic viability phenotypes (Fig EV3E). This suggests that the specificity of mimic toxicity was not defined by the relative presence or absence of the corresponding endogenous miRNA. Gross miRNA mimic dosage effects were also unlikely to account for specificity, as selective responses to miR-210 were preserved across a 10-fold dose response curve (Fig EV3F).

These cumulative observations suggest that distinct miRNA sensitivities are reflecting the presence of distinct acquired molecular vulnerabilities within the PEO1 and PEO4 regulatory frameworks. To help define the nature of these vulnerabilities, we used whole-exome hybridization-capture sequencing (70× average read depth) to estimate cell line-specific somatic mutations and genomic copy number variation together with RNAseq to quantitate relative mRNA expression profiles (Fig EV4A, Datasets EV3, EV4, and EV5, SRA Accession SRP065357). Due to the absence of patient-matched constitutional DNA, we filtered single nucleotide variation (SNV) calls through 16 normal human exomes to quell the detection of

common germline polymorphisms to some extent. Gene-level copy number variation (CNV) was defined using exon read depth at each locus relative to a non-tumorigenic reference cell line. While clearly highly related at the genome level (437 shared SNVs and 2,817 shared focal copy number alterations), 879 cell line-specific SNVs were detected (519 in PEO1 and 360 in PEO4) together with extensive differences in copy number that closely correlated with mRNA expression. We next used these molecular annotations to help inform the biology underlying PEO4-specific miRNA vulnerabilities.

miR-155 and miR-181b mimics were the top-ranked reagents that selectively reduced cell viability in PEO4 cells and were largely innocuous when transfected into immortalized human hepatocytes (IHH), HOSE cells, or human bronchial epithelial cells (HBECs) (Fig 2A). Notably, miR-155 expression is decreased in both ovarian tumors and ovarian cancer cell lines relative to normal tissues and non-tumorigenic cell lines, respectively (Dahiya *et al*, 2008; Zhang *et al*, 2008). The matching phenotypic profiles of miR-181b and miR-155, together with their distinct mRNA targeting sequences, afforded an opportunity to detect predicted targets enriched in overlapping cellular processes. Two independent computational tools, miRPath v1.0 and miRSystem, which determine whether seed-predicted targets of a given miRNA or group of miRNAs are enriched for a particular biological process (using KEGG, Reactome, Biocarta, Pathway Interaction DB), predicted that miR-155 and miR-181b both target insulin signaling (Papadopoulos *et al*, 2009; Lu *et al*, 2012). TargetScan (Lewis *et al*, 2003) predictions of miR-155- and miR-181b-responsive mRNAs included multiple nodes in the AKT and MAPK signaling pathways, both of which are engaged downstream of insulin receptor activation (Fig EV5A). Cell line-specific somatic mutations and copy number alterations predicted reduced AKT pathway responsiveness in PEO4 relative to PEO1 (Fig EV5B). This prediction was validated by examination of serum-induced accumulation of AKT active site phosphorylation (Fig EV5C). Testing in PEO1 cells revealed potent inhibition of AKT activation by these miRNAs that was consistent with mRNA target predictions (Fig 2B). However, direct chemical inhibition of AKT signaling was not sufficient to recapitulate the specificity of mimic expression (Fig EV5D), suggesting that additional activities of miR-155 and miR-181b contribute to selective toxicity. On the other hand, when miR-155 and miR-181b activity was examined in AKT-dependent breast cancer lines, we found a strong correlation of selective sensitivity to these miRs and previously published sensitivities to chemical inhibition of AKT (Garnett *et al*, 2012) (Fig 2C). Thus, while AKT inhibition is not sufficient to account for selective sensitivity of PEO4 cells to miR-155 and miR-181b, these miRNA mimics can effectively target AKT-addicted cancer cells.

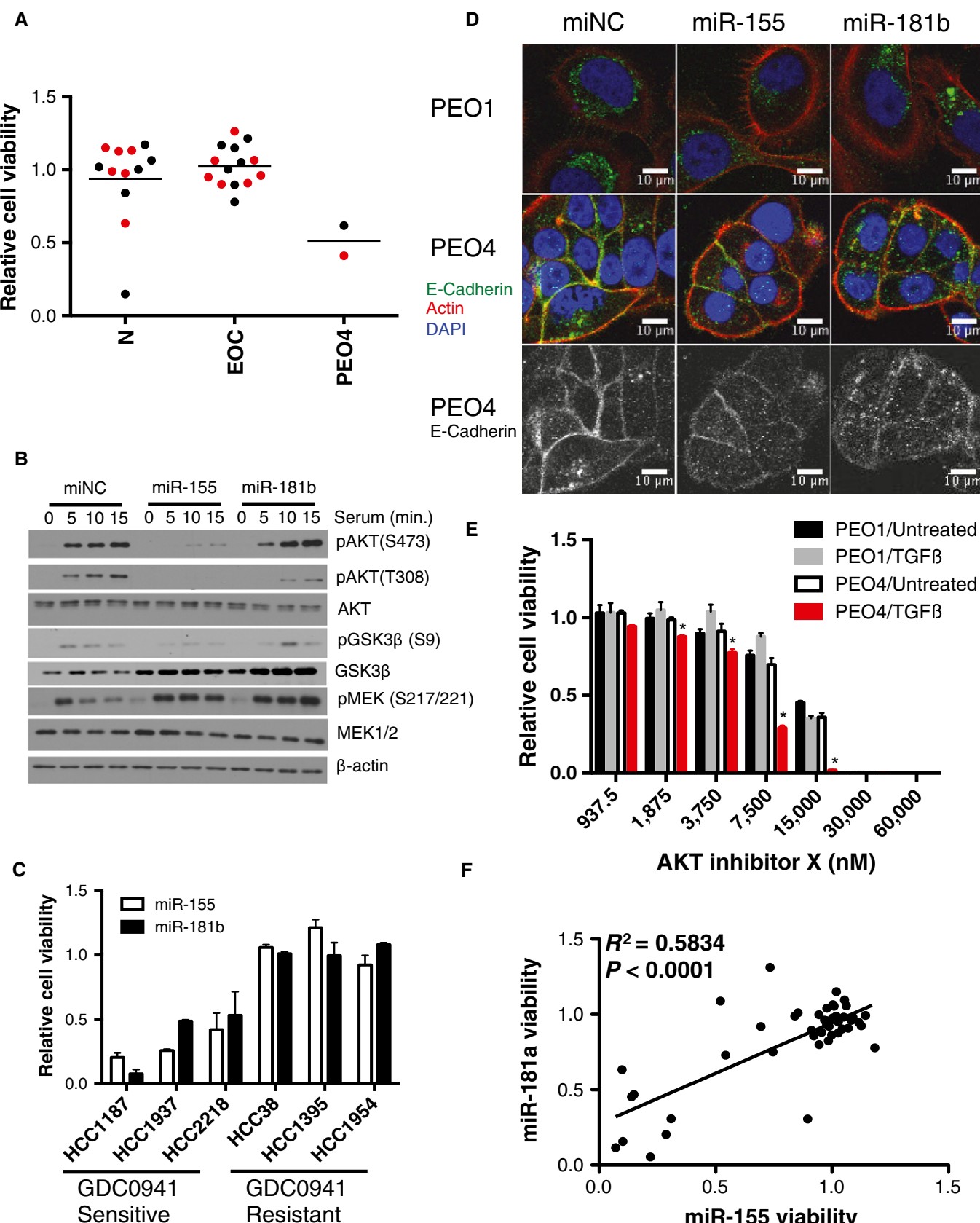

**Figure 2.**

◀ **Figure 2.  Tumor cells with epithelial features are selectively sensitive to miR-155 and miR-181.**

A  The consequence of miR-155 and miR-181b mimics on cell viability, 5 days post-transfection, relative to a negative control miRNA mimic is shown across a panel of normal and ovarian cancer cell lines as indicated. Each data point is the mean of $n = 3$/cell line. In the screen, miR-155 was found to reduce cell viability ($z$-score $< -2$) in one cell line, and TCGA expression data revealed no significant change in expression of this miRNA in ovarian tumors. miR-181 was found to reduce viability ($z$-score $< -2$) in 2 cell lines, and TCGA expression data revealed no significant change in expression of this miRNA in ovarian tumors. N, normal cell lines (IHH, HOSE, HBEC3, HBEC13, HBEC30, and HBEC34); EOC, epithelial ovarian cancer cell lines. miR-155 data points for each cell line are shown in black, while miR-181b data points are displayed in red.

B  Immunoblots indicate suppression of serum-induced AKT phosphorylation at S473 and T308 in response to miR-155 and suppression of T308 phosphorylation in response to miR-181b.

C  miR-155 and miR-181b reduced cell viability in GDC0941-sensitive breast cancer cell lines. Error bars indicate mean $\pm$ SD ($n = 3$).

D  Differential localization of E-cadherin in PEO1 and PEO4 cells 48 h post-transfection with the indicated oligos. Cells were counterstained with phalloidin and DAPI.

E  Stimulation of PEO4 cells with 10 ng/ml TGFβ1 significantly sensitized the cells to AKT inhibition with AKT Inhibitor X. Each point represents the mean of 3 experiments $\pm$ SD and * denotes $P$-value $< 0.05$ by Student's $t$-test.

F  Cell viability upon expression of miR-155 or miR-181a mimics in 41 NSCLC cell lines and 5 human bronchial epithelial cell lines. Each data point is the mean of $n = 3$/cell line. $R^2$ from Pearson correlation. $P$-value calculated from Student's $t$-distribution.

Source data are available online for this figure.

Both miR-155 and miR-181b have recently been identified as effectors of TGFβ induction of epithelial to mesenchymal transition (EMT) downstream of SMAD transcription factor activity (Kong et al, 2008; Cubillos-Ruiz et al, 2012; Johansson et al, 2013; Neel & Lebrun, 2013). Examination of the cell biologic features of the PEO1 and PEO4 cell lines revealed that PEO4 cells formed E-cadherin positive intercellular junctions with a typical epithelial morphology while the PEO1 cells did not. (Fig 2D). Additionally, these discrete junctions were disrupted upon exposure to either miR-155 or miR-181b (Fig 2D). In contrast, PEO1 cells exhibited punctate perinuclear E-cadherin, reminiscent of a mesenchymal phenotype, that was unperturbed by miR-155 or miR-181b (Fig 2D). Taken together, these observations suggested that miR-181b and miR-155 might disrupt epithelial organization, through induction of a mesenchymal transition. Importantly, PEO4 cells were selectively sensitive to the combinatorial effects of TGFβ1 stimulation and AKT inhibition (Fig 2E), suggesting the response to miR-155 and miR-181b can be recapitulated by inhibition of AKT together with induction of mesenchymal transition programs. To further investigate epithelial status as a potential feature specifying cancer cell sensitivity to miR-155 and miR-181, we tested the consequence of miR-155 and miR-181 transfection on cell viability within a panel of 46 non-small cell lung carcinoma-derived cell lines with molecularly defined epithelial and mesenchymal phenotypes (Byers et al, 2013). Toxicity was selective within the panel and the effects of the two miRs were strongly correlated (Fig 2F). Notably, > 70% of the sensitive cell lines (defined by > 50% reduction in cell viability upon miR transfection) expressed epithelial markers. These results, taken together with the observation that non-tumorigenic epithelial cells are resistant to miR-155 and miR-181b (Fig 2A), suggest that epithelial status within an oncogenic regulatory framework is a discriminatory feature specifying sensitivity to miR-155 and miR-181b and that induction of a mesenchymal cell fate by these miRs leads to adverse consequences on cell viability.

## miR-517a targets a common vulnerability in EOC *in vitro* and *in vivo*

miR-517a was one of a small cohort of miRNA mimics with activity in the majority (53%) of the EOC cell panel that was also innocuous in normal ovarian surface epithelial cells, hepatocytes, and human bronchial epithelial cells (Fig 3A). Of note, native miR-517a expression is restricted to the placenta by promoter methylation in other tissues (Yoshitomi et al, 2011; Morales-Prieto et al, 2012). Cancer cell sensitivity to miR-517a expression is not lineage restricted as we also found a large cohort of responsive NSCLC cell lines (Fig 3A). To validate the effects of miR-517a in SKOV3 cells, we assessed the effects of *in vivo* miR-517a delivery in a nude mouse xenograft model using a modified formulation developed for *in vivo* siRNA delivery (Landen et al, 2005). Seven days following intraperitoneal injection of SKOV3 cells, mice were randomly divided and treated twice weekly with miRNA mimics incorporated into DOPC nanoliposomes (i.p. administration) according to the following treatment groups ($n = 8$/group): miR-NC (negative control)/DOPC 400 μg/kg, miR-517a/DOPC 200 μg/kg and miR-517a/DOPC 400 μg/kg. Following a 4-week treatment regimen, mice were sacrificed and necropsied and tumors were harvested. Treatment with miR-517a at either dose significantly reduced both tumor weight and detectable tumor nodules relative to negative control (Fig 3B and C).

To help identify functionally relevant targets of miR-517a, we tested selective sensitivity to miR-517a within a panel of 12 NSCLC cell lines (HBEC30, HCC4017, HCC44, H460, H2122, H2009, H1155, H2073, H1395, H1993, HCC95, H1819, HCC366) for which whole-genome siRNA-toxicity screen results were available from an independent functional genomics effort. The miR-517a sensitivity profile within this panel was then used to identify predicted miR-517a gene targets with matching activity profiles in the siRNA screens. The coatomer complex protein ARCN1 and the ubiquitin thioesterase USP1 were identified in this way (Figs 3D and EV6A), both of which are responsive to miR-517a-mediated suppression of expression in ovarian cancer cell lines (Fig 3E and G). Sensitivity to ARCN1 depletion in the EOC cell line panel also closely phenocopied sensitivity to miR-517a; however, USP1 depletion was selectively toxic to PEO1 cells (Fig 3F). The selective responsiveness of PEO1 cells to USP1 depletion may be a consequence of selective coupling of USP1 to its target protein ID1 and consequent induction of p21 in this background (Williams et al, 2011). These cumulative observations indicate that limiting expression of ARCN1 is a common vulnerability in both epithelial ovarian cancer and non-small cell lung cancer lines that can be artificially perturbed by miR-517a (Fig EV6B).

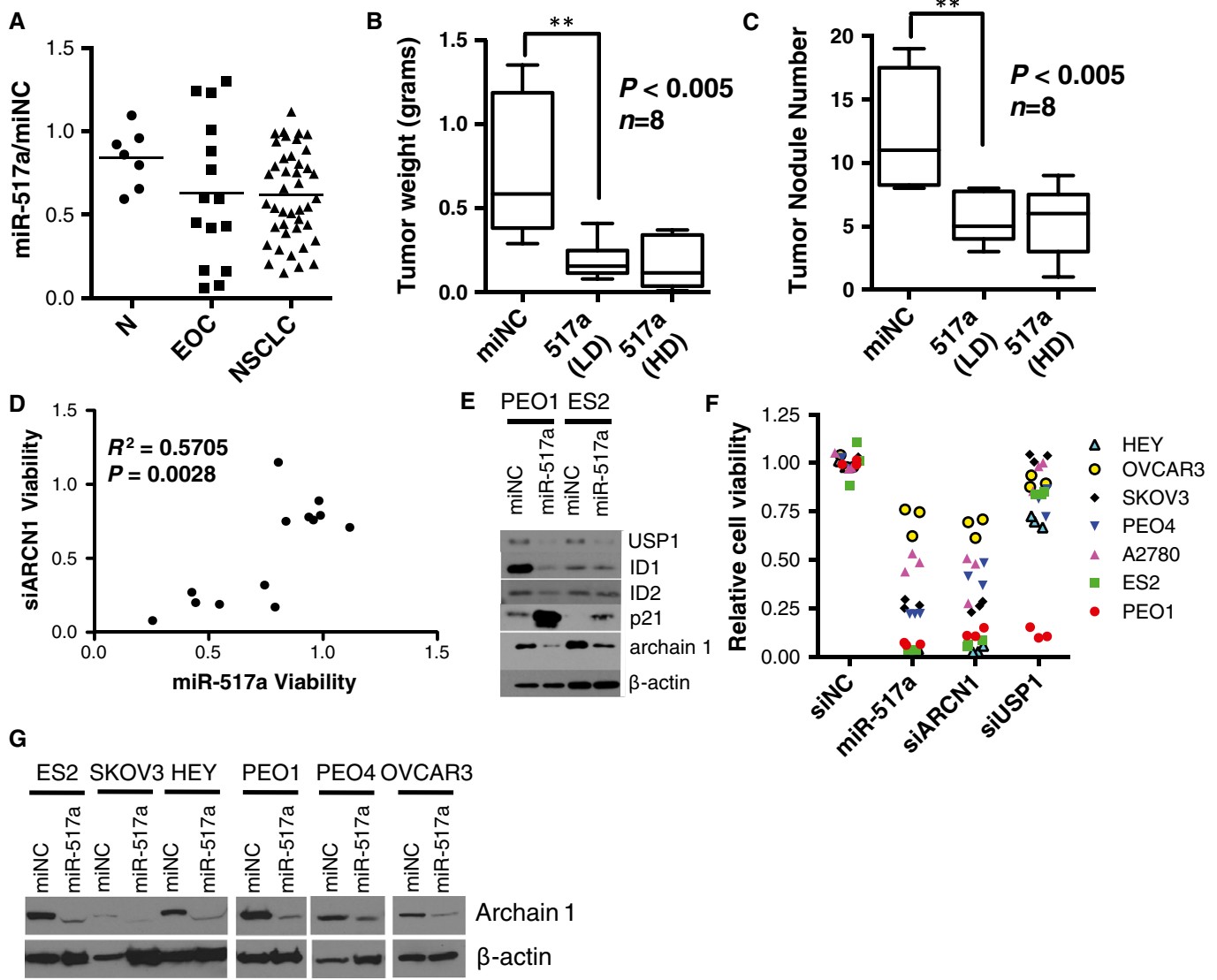

**Figure 3.   miR-517a attacks a common molecular vulnerability in ovarian cancer cells.**

A     Consequence of miR-517a on EOC cell viability and 41 NSCLC cell lines (as in Fig 2A). N, normal cell lines (IHH, HOSE, HBEC3, HBEC13, HBEC30, and HBEC34). miR-517a was found to significantly reduce cell viability ($z$-score $< -2$) in 5 cell lines in the screen, and TCGA expression data revealed no significant change in expression of this miRNA in ovarian tumors.

B, C   *In vivo* delivery of neutral liposome-incorporated miR-517a mimic reduced tumor burden by weight (B) and nodule number (C) in an orthotopic xenograft model using SKOV3 cells. Box-and-whisker plot of tumor weights (B) or tumor nodule number (C) from $n = 8$ mice per condition. **$P$-value from Student's $t$-test. LD, low-dose (200 μg/kg); HD, high-dose (400 μg/kg).

D     Correlation of the consequence of ARCN1 depletion and miR-517a sensitivity in 12 NSCLC cell lines (plot as in Fig 2F). $P$-value calculated from Student's $t$-distribution.

E     Immunoblots indicate miR-517a-induced depletion of ARCN1 and USP1 in the indicated cell lines.

F     Consequence of the indicated siRNAs and miR-517a on the viability of the indicated cell lines.

G     Immunoblots indicate that miR-517a suppresses expression of ARCN1 in multiple miR-517a-sensitive cell lines.

Source data are available online for this figure.

## miR-124 targets SIX4 to release terminal cell differentiation programs

Among the rare cohort of miRNA mimics that inhibited viability in at least 30% of the EOC cell line panel, miR-124 was of particular interest as it was uniformly toxic to all three short-term tumor ascites cultures (HCC5012, HCC5019, and HCC5030, Fig 1A), inert in ovarian surface epithelial cells and hepatocytes (Fig 4A), and is

under-represented on average 3.23-fold in serous ovarian tumors relative to normal tissues (Iorio *et al*, 2007). To help identify the mechanistic basis of ovarian cancer cell sensitivity to miR-124, we first examined the effects of miR-124 on the genomic expression profiles of PEO1 and PEO4 cells (GEO Reference GSE673297 and GSE673298). These lines were chosen given their sensitivity to miR-124 and their association with extensive molecular annotations (Fig EV4A). Seventy-two hours post-transfection with the miR-124

mimic, close to 3,000 genes were detected with altered expression in PEO1 or PEO4 cells (Datasets EV6 and EV9). A major developmental occupation of miR-124 is the promotion of a neuronal differentiation program. The direct suppression of PTBP1 and CTDSP1 by miR-124 can be sufficient to induce the expression of neuronal markers in diverse cell types (Makeyev *et al*, 2007; Visvanathan *et al*, 2007). Consistent with this, we observed miR-124-dependent suppression of PTBP1 and CTDSP1 in PEO1 and PEO4 cells with concomitant induction of the neuronal proteins Tuj1 and MAP2 (Figs 4B and EV7A and B). Extensive evidence indicates that among other described mechanisms of miRNA-mediated regulation (such as inhibition of translation), the direct mRNA targets of miRNAs are depleted upon engagement by a miRNA, with a median downregulation of approximately 2-fold relative to control samples (Lim *et al*, 2005). Remarkably, of the 1,131 miR-124-responsive genes in PEO1 cells, 460 were predicted to be direct seed-driven targets (Lewis *et al*, 2003). Furthermore, there was a strong correlation between

total context score (a numerical evaluator of the likelihood that a predicted target is a *bona fide* target) and the probability of decreased expression of a gene on the microarray. Expressed predicted targets with context scores ≤ −0.2 had a 41% probability of displaying a 2-fold decrease in expression in response to miR-124 (Fig 4C). These observations indicate that supraphysiological concentrations of miRNAs have highly pleiotropic consequences on cellular gene expression programs, and therefore likely influence biological processes via highly combinatorial mechanisms.

Among the cohort of predicted miR-124 targets with the top context scores and robust responsiveness to miR-124 was the homeodomain transcription factor SIX4 together with the eyes absent family (EYA) of SIX protein transcriptional coactivators (Ohto *et al*, 1999) (Fig EV7C). Of note, increased SIX4 expression has been implicated in the suppression of tissue differentiation programs (Yajima *et al*, 2010), and SIX4 is significantly overexpressed in ovarian tumors relative to normal ovarian tissue (Fig EV7D). SIX4

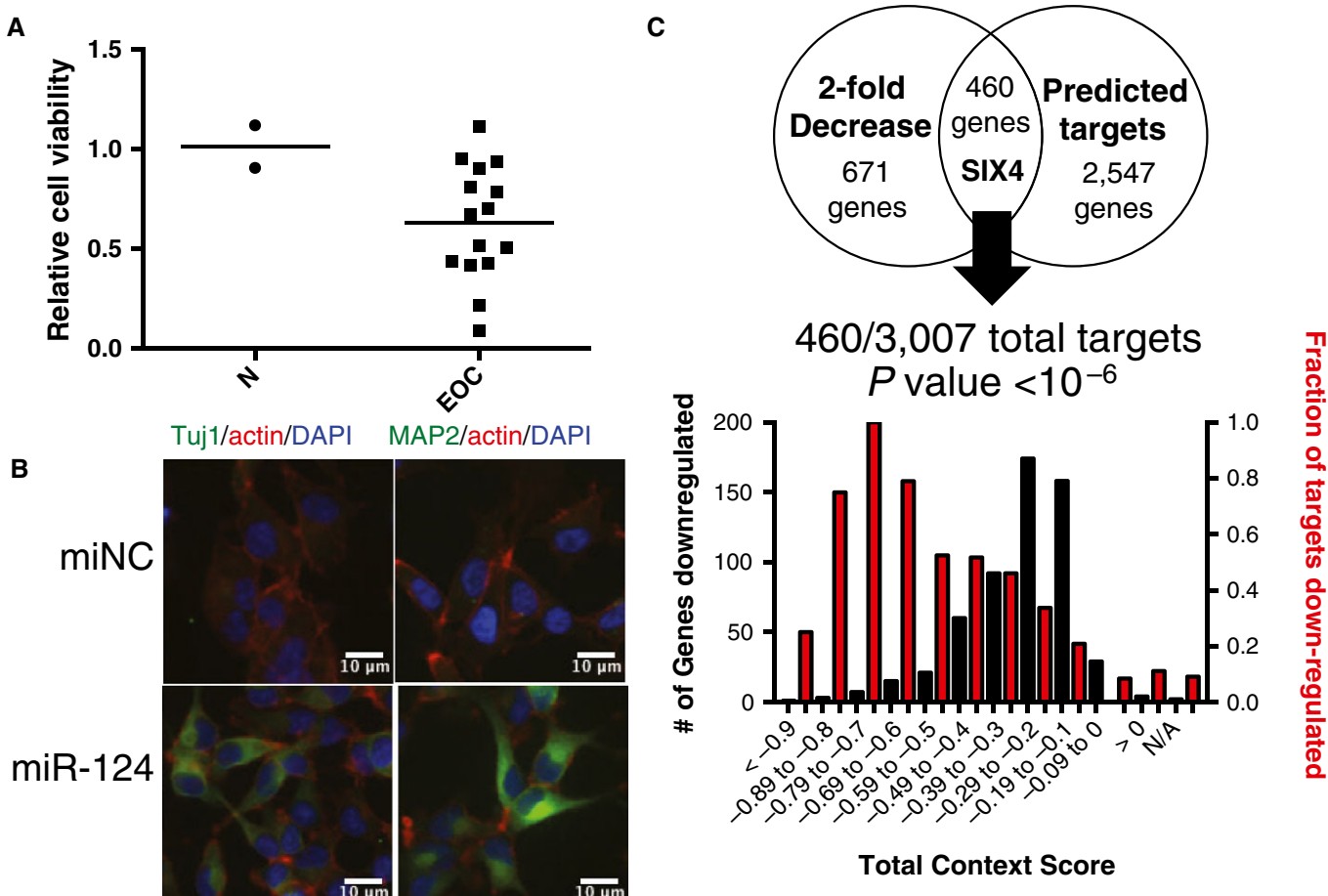

**Figure 4. miR-124-induced reprograming of EOC gene expression profiles.**

A   Consequence of miR-124 on EOC cell viability (as in Fig 2A). N, normal cell lines (IHH and HOSE). In the screen, miR-124 was found to significantly reduce cell viability (*z*-score < −2) in 6 cell lines, and TCGA expression data revealed no significant change in expression of this miRNA in ovarian tumors.

B   miR-124 induced expression of neuronal marker proteins Tuj1 (TUBB3) and MAP2 in ES2 cells 48 h post-transfection. Cells were counterstained with DAPI and phalloidin.

C   miR-124-responsive genes in PEO1 cells were enriched for TargetScan predicted targets. *P*-value from hypergeometric distribution.

Source data are available online for this figure.

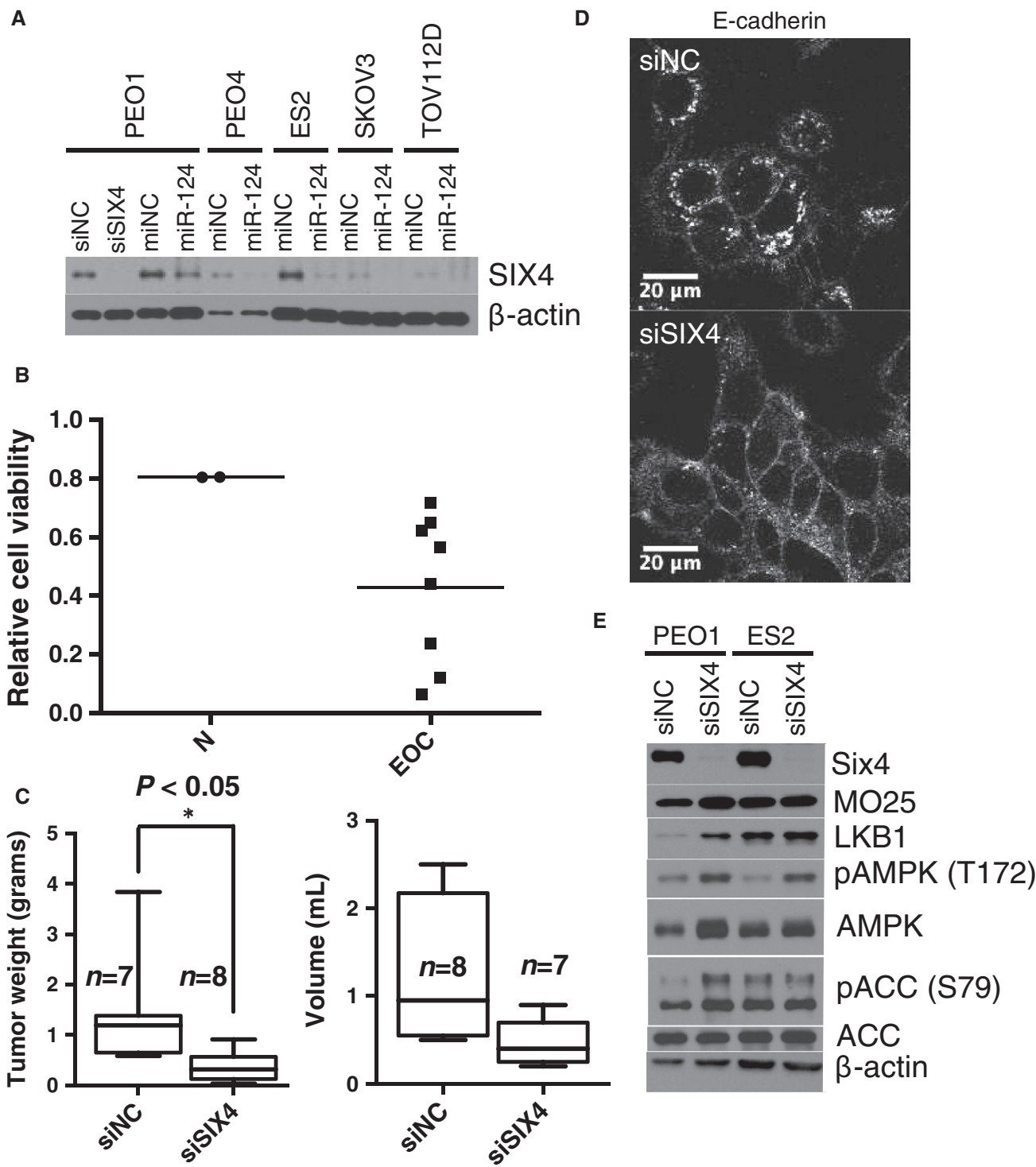

**Figure 5.  Selective addiction of EOC to the SIX4 homeobox transcription factor.**

A   Immunoblots indicate miR-124-induced SIX4 depletion. siSIX4 is shown as an antibody control.

B   Consequence of SIX4 depletion on EOC cell viability (as in Fig 2A, N, normal cell lines: HOSE, HBEC30).

C   *In vivo* knockdown of SIX4 using neutral liposome-incorporated siRNA reduced tumor burden and ascites volume in an orthotopic xenograft mouse model. Box-and-whisker plot of tumor weight (left panel) or ascites volume (right panel) from *n* = 7 or 8 mice per condition as indicated. *P-value from Student's *t*-test.

D   Consequence of SIX4 depletion in PEO1 cells on E-cadherin plasma membrane accumulation 48 h after transfection. Cells were counterstained with phalloidin and DAPI.

E   Immunoblots indicate consequence of SIX4 depletion on LKB1 accumulation and AMPK pathway activation.

Source data are available online for this figure.

protein levels decreased upon miR-124 expression in multiple ovarian cancer cell lines (Fig 5A), and siRNA-mediated SIX4 depletion was selectively toxic within the EOC cell line panel (Figs 5B and EV7E). Expression of a SIX4 gene lacking its 3′-UTR, and thus miR-124 target sites, partially abrogated miR-124-induced toxicity in PEO1 cells (Fig EV7F). Tumor responsiveness to SIX4 depletion *in vivo* was modeled using ES2 cell xenografts. Tumors were allowed to form in the peritoneum over the course of 7 days before delivery of DOPC neutral liposomes incorporating SIX4 siRNA ($N = 8$ mice) or negative control siRNA ($N = 7$ mice). Following a 4-week treatment regimen, SIX4 siRNA-treated animals displayed significantly decreased total tumor weight and concomitantly reduced ascites (Fig 5C).

To help elaborate the mechanistic consequences of SIX4 inactivation, we examined the effect of SIX4 depletion on PEO1 and PEO4 genomic expression profiles (Datasets EV7 and EV10). Reduced expression of multiple cyclins correlated with reduced cell cycle progression and accumulation of cells in G1 (Fig EV7G and H). In addition, we observed significant induction of STRADB (Fig EV7I). STRADB stabilizes LKB1 to promote activation of AMPK. One consequence of STRAD-induced AMPK signaling is mesenchymal to epithelial transition accompanied by the formation of E-cadherin-positive adherens junctions (Zhang *et al*, 2006). We observed a marked increase in E-cadherin-positive cell junctions, 48 h post-SIX4 depletion in PEO1 cells, suggesting SIX4 depletion was sufficient to re-establish an epithelial phenotype in these cells (Fig 5D). This was accompanied by LKB1 accumulation, and AMPK pathway activation (Fig 5E). These observations suggest that SIX4 expression in ovarian cancer cells both promotes cell cycle progression and inhibits tumor-suppressive AMPK pathway activity (Fig EV7J).

## Discussion

Accumulating evidence indicates an association of loss of microRNA biogenesis with the development of ovarian cancer. Here, we have employed micro-RNA mimic toxicity screens to evaluate the sensitivity of ovarian epithelial cancer cell lines to individual mature miRNAs. A striking feature of the resulting toxicity profile was the robust but idiosyncratic vulnerabilities displayed within the cell line panel. Modeling of these idiosyncratic responses with patient data, and in other tumor lineages, suggests they are reflective of fractional representation of diverse phenotypes in EOC. This behavior is consistent with a common sensitivity of EOC to miRNA production, which, however, is driven by private miRNA species as a consequence of the diverse molecular etiologies found in this disease.

Despite the preponderance of idiosyncratic activity, a small cohort of commonly toxic mimics was identified. The mature sequence from hsa-miR-517a inhibited viability in 53% of the EOC lines tested and was innocuous in telomerase-immortalized ovarian epithelial cells and hepatocytes. ARCN1 was identified as a miR-517a target associated with miR-517a sensitivity. This protein is part of the COPI complex, which otherwise supports retrograde transport, acidification of autophagolysosomes, and is a CDC42 effector required for CDC42 transformation (Wu *et al*, 2000; Razi *et al*, 2009; Huotari & Helenius, 2011). Notably, direct ARCN1 depletion phenocopied the miR-517a toxicity profile in both ovarian and lung cancer cell lines, suggesting ARCN1 function is limiting in a large cohort of cancer cells and can be artificially targeted by miR-517a.

An additional common vulnerability was identified with miR-124, a microRNA that has been extensively studied in relation to its role in specification of neuronal cell fate. miR-124 expression is under-represented in multiple tumor types as compared to corresponding normal tissues, including serous ovarian adenocarcinoma. We found that miR-124 has an extensive mRNA target space in ovarian epithelial cancer cells; however, suppression of the homeobox transcription factor SIX4 was sufficient to mimic the consequences of miR-124 on ovarian cancer cell proliferation. SIX4 is overexpressed in ovarian tumors relative to normal tissue and maintains cancer cell proliferation, at least in part, by supporting cyclin gene expression and suppressing AMPK pathway activation. SIX4 depletion resulted in induction of cell differentiation programs concomitant with terminal cell cycle arrest. Systemic delivery of siRNA targeting SIX4 effectively inhibited xenograft tumor growth, nominating SIX4 and/or SIX4 target genes as an ovarian cancer intervention target.

Deleterious mobilization of cell differentiation programs was a shared mechanism underpinning the antitumorigenic activity of many of the miRs studied here. miR-155, miR-181, miR-517a, and miR-124 have been implicated in the differentiation of hematopoietic cells, adipocytes, osteoblasts, embryoid tissue, and neurons (Chen *et al*, 2004, 2013b; Bhushan *et al*, 2013; Eguchi *et al*, 2013). As a class, miRs are intimately associated with differentiation programs, exhibit tissue-specific expression, and can be sufficient to induce anomalous expression profiles in heterologous contexts that are reminiscent of the miRNA's tissue of origin (Lagos-Quintana *et al*, 2002; Lim *et al*, 2005; Landgraf *et al*, 2007). Thus, the common loss of miRNA expression and maturation in ovarian cancer cells might serve to deflect anomalous engagement of cellular differentiation programs in response to oncogene activation, offering differentiation therapy for consideration as a potential treatment modality for ovarian cancer. Though distinct from the solid tumor context, perhaps the most well-known example of differentiation therapy is the treatment of acute promyelocytic leukemias (APMLs) with all *trans*-retinoic acid (ATRA). APML is associated with relatively simple genetic background, where almost all patients share a distinct chromosomal rearrangement that confers ATRA sensitivity (Larson *et al*, 1984). The observations presented here suggest that vulnerability to differentiation programs may also lie at the nexus of the diverse oncogenotypes associated with EOC, and therapeutic agents modeled on miR-155/181, miR-517a, and miR-124 may therefore offer intervention opportunities for large cohorts of the ovarian cancer patient population.

## Materials and Methods

### Cell culture

PEO1 and PEO4 cells were generously provided by Hani Gabra (University of Edinburgh Cancer Research Center). ES2 and TOV112D cell lines were purchased from the ATCC. SKOV3, Hey, OC316, OVCAR3, A2780, EFO21, EFO27, OAW42, and UPN251 cell lines were generously provided by Dr. Robert Bast (MD Anderson Cancer Center). IHH cells were generously provided by Dr. John

Abrams (UT Southwestern). HOSE cells were generously provided Dr. William Hahn (Dana Farber Cancer Institute, Harvard). PEO1, PEO4, SKOV3, ES2, TOV112D, Hey, OC316, OVCAR3, A2780, EFO21, EFO27, OAW42, and UPN251 cell lines were grown in RPMI medium with L-glutamine and 25 mM HEPES (Gibco) supplemented with 10% fetal bovine serum (Atlanta Biologicals) and 100 U/ml penicillin and 100 μg/ml streptomycin (Gibco). HCC5012, HCC5030, and HCC5019 were grown in ACL4 medium supplemented with 5% fetal bovine serum (Atlanta Biologicals). HOSE cells were grown in Keratinocyte Serum Free Medium with supplements (Gibco) and 100 U/ml penicillin and 100 μg/ml streptomycin (Gibco). IHH cells were grown in DMEM medium (Gibco) supplemented with 10% fetal bovine serum (Atlanta Biologicals) and 100 U/ml penicillin and 100 μg/ml streptomycin (Gibco). Upon receipt, the identity of all cell lines was confirmed via short-tandem repeat DNA fingerprinting that was then compared to a database of known samples. Transfections were performed as described in the miRNA mimic screens.

### miRNA mimic screens and transfection

All cell lines were screened using a collection of miRNAs consisting of the intersection of two Dharmacon miRIDIAN Mimic Libraries inclusive of all miRNAs annotated in miRBase 8.0 and miRBase 10.1, respectively (www.mirbase.org). Screens were performed similarly to those previously described with slight modifications (Whitehurst *et al*, 2007; Ganesan *et al*, 2008). The transfection conditions for each cell line were optimized by varying transfection reagent and cell number plated while keeping the mass of RNA transfected constant to ensure minimal transfection-related toxicity and maximal transfection efficiency. Transfection-related toxicity was assessed by transiently transfecting in a negative control mimic and normalizing cell viability values to non-transfected wells 120 h post-transfection using the same Cell Titer Glo endpoint as the screen. Transfection efficiency and dynamic range were assessed in a similar manner by transiently transfecting each cell line with the pan-toxic siRNA siUBB and then using the Cell Titer Glo endpoint assay 120 h post-transfection. 10 picomoles of miRNA mimic per well were delivered in 30 μl serum free medium to 96-well microtiter plates using a Biomek FX robotic liquid handler (Beckman Coulter). A total of 9.8 μl of serum free medium containing 0.2 μl of RNAiMAX transfection reagent was then added to each well using a TiterTek multidrop (except for PEO4 where 0.2 μl of DharmaFECT 3 was used) followed by a 20- to 30-min incubation at room temperature. Single-cell suspensions were then delivered to each well using a TiterTek multidrop to a total volume of 150 μl. The total number of cells per well varied according to the optimal transfection conditions for each cell line as follows: PEO1 (10,000), PEO4 (10,000), SKOV3 (5,000), A2780 (20,000), OC316 (5,000), EFO21 (5,000), Hey (5,000), OVCAR3 (12,500), IGROV1 (12,500), EFO27 (15,000), UPN251 (10,000), OAW42 (10,000), CAOV3 (5,000), HCC5012 (3,000), HCC5030 (5,000), and HCC5019 (5,000). All cells were plated in the media in which they were cultured as described above. Plates were then centrifuged at 500 rpm for 1 min and incubated in a 37°C/5% CO$_2$ incubator. Seventy-two hours after plating cells, 50 μl of fresh media was added to each well using a TiterTek multidrop. About 120 h after plating, 15 μl of CellTiter-Glo Reagent (Promega) was added to each

well and incubated per manufacturer's protocol. Luminescence values for each well were then recorded using an Envision Plate Reader (Perkin Elmer). Each transfection was performed in triplicate. Raw luminescence values for each well were normalized to allow for comparisons from well to well and plate to plate. Each well in a row was normalized to the median value for the row. The *z*-score for each well was then derived using siMACRO macro for Excel (Singh *et al*, 2013). Transfection of siRNAs was performed as described above using 10 pmoles of a pooled siRNA instead of miRNA mimic. HOSE and IHH transfection were performed as above plating 10,000 cells per well and 2,000 cells per well, respectively. miRNA mimic and siRNA oligo sequences can be found in Table EV2.

To evaluate the capacity of ectopic miRNA target expression to rescue miRNA mimic toxicity, the relevant oligo was transfected into cells as described above. 24 h after mimic transfection, relevant plasmids expressing miRNA target cDNA or an empty vector was transfected into cells via PolyJet (Signagen). Twenty-four hours later, fresh medium with 0.3 μg/ml puromycin was added to each well. The experiment was terminated 120 h after the mimic transfection and cell viabilities were assessed using Cell Titer Glo as above.

### Data processing

Clustering analysis was performed with the affinity propagation clustering (APC) algorithm using the "apcluster" package in R. APC is a deterministic clustering method which identifies the number of clusters, and cluster "exemplars" (i.e., the cluster centroid or the data point that is the best representative of all the other data points within that cluster) entirely from the data (Frey & Dueck, 2007), giving it an advantage over non-deterministic methods subject to a biased randomized initialization step, such as Hierarchical Clustering, or methods in which the number of clusters has to be pre-specified, such as k-means clustering.

Affinity propagation clustering performs clustering by passing messages between the data points. It takes as input a square matrix representing pairwise similarity measures between all data points (either Euclidean distance or Pearson correlations). The algorithm views each data point as a node in a network and is initialized by connecting all the nodes together where edges between nodes are proportional to Pearson correlations. The algorithm then iteratively transmits messages along the edges, pruning edges with each iteration, until a set of clusters and exemplars emerges.

Two real-valued messages are passed between nodes. The "responsibility" message computes how well suited it is for point $i$ to choose point $k$ as an exemplar, given all the other candidate exemplars, $k'$, and is updated by:

$$r(i,k) \leftarrow s(i,k) - \max_{k' \, st \, k' \neq k}\{a(i,k') + s(i,k')\}$$

The availability message, $a(i, k)$, computes how appropriate it is for point $i$ to select point $k$ as an exemplar, taking into account all the other points for which $k$ is an exemplar, $i'$, and is given by:

$$a(i,k) \leftarrow \min\left\{0, r(k,k) + \sum_{i' \, st \, i' \notin \{i,k\}} \max(0, r(i',k)\right\}$$

In the above equation, $a(i, k)$ is set to the self responsibility, $r(k, k)$, plus the sum of the positive responsibilities candidate

$k$ receives from other points. The entire sum is thresholded at 0, and a negative availability indicates that it is inappropriate for point $i$ to choose point $k$ as an exemplar and the tie is severed. The self-availability, $a(k, k)$, reflects the accumulated evidence that point $k$ is an exemplar and is updated with the following rule, which reflects the evidence that $k$ is an exemplar based on the positive responsibilities sent to $k$ from all points, and is updated by:

$$a(k, k) \leftarrow \sum_{i'\,st\,i' \notin \{i,k\}} \max(0, r(i', k)$$

In the first iteration, all points are considered equally likely to be candidate exemplars, and $a(i, k)$ is set to 0 and $s(i, k)$ is set to the input similarity measure between points $i$ and $k$. The above rules are then iteratively updated until a clear, stable set of clusters and exemplars emerges.

In our implementation of the algorithm, we first ran the algorithm to identify an initial set of exemplars and clusters from the data. The exemplars were then clustered together and this procedure was repeated until no more clusters emerged to identify a hierarchical structure of clusters. Networks were drawn with cytoscape (Shannon *et al*, 2003).

RNASeq data for 7 cell lines collected from the CCLE (Barretina *et al*, 2012) (http://www.broadinstitute.org/ccle) and 2 cell lines from this study (Dataset EV5) were combined. The RNASeq data was first filtered to contain only the top 20% of the most highly variant genes (2,686 genes total). The cell lines were then clustered using hierarchical APC clustering (described above) based on a Euclidean distance metric.

The 400 microRNAs were clustered together based on their *z*-scores across 16 cell lines using a Euclidean distance metric, and the 16 cell lines were clustered together based on their *z*-scores across 400 miRs using a Pearson correlation metric.

For each microRNA mimic, a standard deviation value was calculated for viability across the panel of cell lines. For all miRNAs in the same seed family, a standard deviation value was calculated for viability across the panel of 16 cell lines. Lastly, a within-replicate standard deviation value was calculated. A kernel density estimation was fit to each of the three standard deviation distributions and plotted.

Identification of potentially active miR-517a targets was achieved as follows. A total of 52 miR-517a target genes were predicted by TargetScan 6.0 (www.targetscan.org). These predicted targets were then filtered for activity (relative cell viability < 0.5, $N = 10$) in a miR-517a-sensitive cell line, H2122, that had also previously undergone a genomewide siRNA toxicity screen. Toxicity in response to depletion of each of these 10 predicted targets, with two independent siRNA pools, was then assessed for correlation with toxicity upon transfection with miR-517a mimic in a panel of 13 NSCLC lines. Only 2 predicted targets, USP1 and ARCN1, demonstrated a positive correlation with miR-517a ($R^2 > 0.35$) using siRNA oligos from both Dharmacon and Ambion.

Statistical analysis of data was performed using mainly nonparametric tests. However, parametric tests (i.e. *t*-test) were used when performed on data with a normal distribution. Standard deviation from the mean was used to assess variance within a data set and to ensure variance between data sets was similar.

### Derivation of new cell lines

Because cell lines may lose phenotypic properties during long term culture (Gillet *et al*, 2013), we derived new ovarian cell lines and cryopreserved working stocks from them at early time points after culture initiation (3–8 months). Malignant ascites fluid from patients with untreated high-grade papillary ovarian carcinoma were treated with hemolysis agents to remove red blood cells and then enriched for tumor cells by a series of differential low speed centrifugations and differential attachment to culture dishes, and plated into culture flasks with ACL4 medium and 5% fetal bovine serum. Cell lines were cryopreserved as soon as continuous growth for 3–4 passages occurred. Cell lines were epithelial in morphology and over 80% of the cells expressed epithelial cell adhesion molecule (EPCAM) (van der Gun *et al*, 2010). Derivation of cell lines was approved by the UT Southwestern IRB, and informed consent was obtained from all individuals involved.

### Antibodies and compounds

Antibodies were purchased from Cell Signaling (pAKT (T308) #2965, pAKT (S473) #4060, panAKT #4691, pGSK3β (S9) #9327, GSK3β #9315, p27 #3686, TSC2 #3635, pMEK (S217/221) #9121, MEK1/2 #9122, ERK1/2 #9102, pERK (T202/Y204) #4370, SMAD4 #9515, pSMAD3 (S423/425) #9520, SMAD3 #9523, SMAD2 #5339, USP1 #8033, p21 #2947, LKB1 #3050, pAMPK (T172) #2535, AMPK #2532, pACC (S79) #3661, ACC #3676, Claudin-1 #4933, Vimentin #5741), Sigma (β-actin #A1978), Millipore (pSMAD2 (S465/467) #AB3849), Santa Cruz (ID1 #SC-488 and ID2 #SC-489), Novus (SIX4 #51804-M09 and ARCN1 #NBP1-32377), and Epitomics (MO25 #2027-1). HRP-conjugated secondary antibodies were purchased from Jackson Immunolaboratories, and ECL reagents were purchased from ThermoScientific. AKT Inhibitor X was purchased from EMD Millipore (Cat #124039). LY2940092 was purchased from Sigma (Cat #L9908). TGFβ1 protein was purchased from Peprotech (Cat #100-21).

### Immunofluorescence

siRNAs and miRNAs were delivered by reverse transfection as described above and seeded on coverslips. After 48 h, cells were washed with PBS and fixed with 4% methanol-free formaldehyde (Fisher). Cells were then permeabilized with 0.5% Triton X-100 in PBS. Manufacturer's protocols were then followed for blocking and incubation with primary antibodies except all primary antibodies were used at 1:100 dilution. Primary antibodies were purchased from BD Biosciences (E-cadherin #610181) and Covance (Tuj1 #MMS-435P and MAP2 #SMI-52R). Cells were then incubated with Alexa 488-conjugated secondary antibodies (Invitrogen) at 1:500 for 1 h at room temperature. Coverslips were then stained with Texas Red-X phalloidin per manufacturer's protocol (Invitrogen #T7471). Coverslips were then mounted onto slides using Vectashield (Vector Labs) mounting medium containing DAPI. Slides were imaged using the Leica TCS SP5 confocal microscope (Leica Micro-systems, CMS GMBH) with custom software (Leica Micro-systems LAS AF) using a sequential 3-channel scan. All images were captured using the same electronic settings. Images were then imported in ImageJ (http://rsb.info.nih.gov) using the LOCI Bio-formats plug-in (University of

Wisconsin, Madison). Tuj1 and MAP2 staining was imaged using a Zeiss Axioplan 2E with a Hamamatsu monochrome digital camera. OpenLab (Improvision) software was used for image acquisition on the Zeiss microscope. All image processing was performed in ImageJ.

### Animals, orthotopic *in vivo* model and tissue processing

Female athymic nude mice were purchased from the National Cancer Institute, Frederick Cancer Research and Development Center (Frederick, MD). These animals were cared for according to the guidelines set forth by the American Association for Accreditation of Laboratory Animal Care and the U.S. Public Health Service policy on Human Care and Use of Laboratory Animals. All mouse studies were approved and supervised by the M.D. Anderson Cancer Center Institutional Animal Care and Use Committee. All animals used were between 8 and 12 weeks of age at the time of injection. A standard power calculation for detection of a 50% effect size was used to determine sample size. For the miR-517a experiment, SKOV3ip1 cells were trypsinized, washed, and resuspended in Hanks' balanced salt solution (Gibco, Carlsbad, CA) and injected intraperitoneally into mice (SKOV3ip1: $1 \times 10^6$ cells/animal). Similarly, for the SIX4 siRNA experiment, ES2 cells ($2.5 \times 10^5$ cells/animal) were prepared and injected intraperitoneally. For both experiments, 7 days after the tumor cell injection, mice were randomly divided and treated with oligonucleotides incorporated in neutral nanoliposomes (intraperitoneal [IP] administration). For the miR-517a experiment, mice were randomized to the following three groups ($n = 10$/group): negative control miRNA/DOPC or miR-517a/DOPC at either 200 µg/kg or 400 µg/kg. For the SIX4 experiment, mice were randomized to the following two groups ($n = 10$/group): negative control siRNA or SIX4 siRNA. For both experiments, twice weekly treatments continued for 4–5 weeks at which point, all mice in the experiment were sacrificed and necropsied, and tumors were harvested. Tumor weights, number, and location of tumor nodules were recorded. Tumor tissue was either fixed in formalin for paraffin embedding, frozen in optimal cutting temperature (OCT) media to prepare frozen slides, or snap-frozen for lysate preparation. Researchers were not blinded to treatment group.

### Liposomal preparation

miRNA for *in vivo* delivery was incorporated into DOPC as previously described (Landen *et al*, 2005). DOPC and miRNA were mixed in the presence of excess tertiary butanol at a ratio of 1:10 (w/w) miRNA/DOPC. Tween 20 was added to the mixture in a ratio of 1:19 Tween 20:miRNA/DOPC. The mixture was vortexed, frozen in an acetone/dry ice bath, and lyophilized. Before *in vivo* administration, this preparation was hydrated with PBS at room temperature at a concentration of 200 µg/kg per injection.

### Exome sequencing

For each cell line, 5 µg of genomic DNA was isolated for whole-exome capture sequencing. In brief, DNA was fragmented (150–250 bp) and ligated to the paired-end adaptors. The adaptor-ligated fragments were then amplified by PCR and purified. Exon-containing fragments in these libraries were hybridized to the SureSelect Human All Exon Kit from Agilent technologies. This kit

targets 165,637 exons (~18,003 genes), totaling approximately 38 Mb of genomic DNA. The hybridized fragments were then captured using streptavidin-coated magnetic beads and amplified and each sample was sequenced in the UT Southwestern Genomics Core Facility in two lanes of an Illumina GAIIx using a standard 75-bp paired-end protocol. The image analysis and base calling were performed using the Illumina pipeline with default settings. Prior to analysis, duplicate reads (multiple fragments from the same amplicon), identified on the basis of having the same start position for both end reads, were removed from the sequence analysis. For copy number analysis, a total of 88 million read pairs ($2 \times 74$ bp) for PEO1 and 89 million read pairs for PEO4 passed QC, and 148 million reads from each of the two lines were uniquely aligned to NCBI human genome build 37 by Bowtie 0.12.5 (Langmead *et al*, 2009) allowing up to 2 mismatches per read. Genomewide copy number variation was analyzed for the pair of cell lines separately using completely unrelated normal tissue data obtained by others with the same exome capture kit as a reference for normalization.

### RNA sequencing

RNA was isolated in triplicates from PE01 and PE04 cell lines using the RNeasy Kit (Qiagen), and the quality of RNA was checked using a Bioanalyser. From each sample, 5 µg of RNA was used to perform RNA-Seq using the Illumina mRNA Sequencing Sample Preparation Guide (Illumina, Cat # RS-930-1001). First poly-A containing mRNA was purified using poly-T oligo-attached magnetic beads and then fragmented using divalent cations under elevated temperatures. Then, the first and second strand cDNA was synthesized using random primers, end-repaired, adenylated, and ligated with paired-end adapters. The products were then purified and enriched with PCR to create the final cDNA library. The library from each sample was sequenced in a single lane of an Illumina GAIIx using a standard 40-bp paired-end protocol. Reads were mapped to the UCSC *Homo sapiens* reference genome hg19 and their relative expression values were calculated in RPKM using CLC Biosystems Genomic Workbench software.

### Whole-genome expression microarrays

RNA was isolated from cells 72 h post-transfection using an RNeasy kit (Qiagen). Illumina HumanWG-6 V4 BeadChip (Illumina, Inc.) human whole-genome expression arrays, which contain 47,231 probes on each array, were used. Each RNA sample was amplified by Ambion TotalPrep RNA amplification kit with biotin UTP (Enzo) labeling, using 500 ng of total RNA. The Ambion Illumina RNA amplification kit uses T7 oligo(dT) primer to generate single-stranded cDNA followed by a second-strand synthesis to generate double-stranded cDNA which is then column-purified. *In vitro* transcription with T7 RNA polymerase generated biotin-labeled cRNA. The cRNA was then column-purified, checked for size and yield using the Bio-Rad Experion system, and then 1.5 µg of cRNA was hybridized to each array using standard Illumina protocols with streptavidin-Cy3 (Amersham) being used for detection. Slides were scanned and fluorescence intensity captured using an Illumina BeadStation. Expression values from were extracted using BeadStudio v3.3. The data were background subtracted and quantile-normalized using the MBCB algorithm (Ding *et al*, 2008; Allen *et al*,

2009; Xie *et al*, 2009). The labeling and hybridization of miRNA was performed according to the miRNA Expression Profiling Assay Guide from Illumina Inc. Briefly, 1 μg of total RNA was polyadenylated. The RNA was then converted to cDNA with oligo-dT primers and a universal PCR sequence. The cDNA was captured by using a pool of oligonucleotides that have miRNA-specific sequences. The captured cDNA containing miRNA sequences were amplified by PCR. The strands that have complementary sequences to the probes on the array were labeled with Cy3. A single-stranded PCR product that is fluorescent-labeled was finally prepared and hybridized on Illumina Universal-12 BeadChip Human MI v2 miRNA arrays. After hybridization, the array slides were scanned on an Illumina Beadstation. Signal intensities of microarrays were summarized using BeadStudio v3.3 (Illumina, Inc). Background subtraction and quantile normalization were performed using the MBCB algorithm.

## Clinical outcomes and associations

We downloaded and analyzed data publicly available from the Cancer Genome Atlas Project (TCGA; http://tcga-data.nci.nih.gov/) for patients with ovarian serous cystadenocarcinoma. Level 3 IlluminaHiSeq miRNASeq and Agilent MicroRNA microarray data were used for miRNA expression. For miRNASeq data, we derived from the "isoform_quantification" files containing the "reads_per_million_miRNA_mapped" values for mature forms for each microRNA. Survival analyses were performed in R (version 2.14.2), and the statistical significance was defined as a *P*-value less than 0.05. The Log-rank test was employed to determine the relationship between expression and overall survival, and the Kaplan–Meier method was used to generate survival curves. We randomly split the entire population into training/validation cohorts (2/3, 1/3). For each miRNA, we checked for a relation with the survival as follows. In both cohorts, patients were divided into percentiles according to the miRNA expression. Using the training set, we considered any cutoff between the $25^{th}$ and $75^{th}$ percentile that significantly split the samples and verified the statistical significance in the validation set.

For SIX4 expression analysis, we used the gene expression data run by UNC on Agilent Expression 244K microarrays measuring 17,814 genes. The data involve 598 samples, of which 36 samples were run by one batch and the remaining 563 were run by the other batch. We performed principal component analysis (PCA) on the combined expression data and found no obvious batch effect. Of the 598 samples, 573 are primary solid tumors, 17 are recurrent solid tumors and 8 are normal solid tissue samples. We extracted the expression levels of SIX4 from all samples and generated a box plot. The data processing and statistical analyses were performed in R (R-Core-Team, 2012).

## Cell cycle and growth assays

For evaluation of caspase 3/7 activation in cells, we transfected cells as above except cells were plated in a total volume of 100 μl. After 48 h, cells were incubated with 50 μl of CaspaseGlo 3/7 reagent (Promega) per manufacturer's protocol. Luminescence values were read on a PheraStar plate reader (BMG LabTech), and raw luminescence values were normalized to a negative control siRNA contained on each plate. BrdU incorporation assays were performed by incubating cells with 10 μM BrdU for 4 h, 48 h after transfection as described above. After incubation, cells were fixed with 3.7% paraformaldehyde. DNA was denatured using 0.5 N HCl, and cells were stained with anti-BrdU antibody conjugated to Alexa-488 (Invitrogen) per manufacturer's instructions. Hoechst dye (Invitrogen) with then diluted in PBS per manufacturer's protocol and added to each well. Plates were read using a BD Pathway 855 microscope (BD Biosciences). Using AttoVision software (BD Biosciences), cells were segmented and Hoechst-positive and FITC-positive nuclei were automatically counted. Cutoffs to segregate positive nuclei were empirically determined and constant for the entire plate. DNA content analysis was performed 48 h after transfection. Cells were stained with propidium iodide (PI) using a PI/RNase Buffer (BD Biosciences) following manufacturer's protocol. Samples were run on a FACSCalibur flow cytometer and acquired with CellQuest Pro (Becton Dickinson, San Jose, CA).

Samples were analyzed with FlowJo (Treestar). Gating was performed to gate out dead cells and doublets and then Dean/Jet/Fox modeling was applied.

## Plasmids

pRK5-Myc-SIX4 contains the human SIX4 coding sequence (nucleotides 1–2,283, from pANT7-SIX4-cGST (dnasu.org)) inserted in the BamHI/XbaI sites of pRK5-Myc (Clontech). pRK5-Myc-ARCN1 contains the human ARCN1 coding sequence (nucleotides 1–1,536 plus "TACCAAGAAGAGGGAGC", a 17 bp 3′-UTR fragment immediately downstream the ARCN1 cDNA from MGC Human ARCN1 Sequence-verified cDNA (Thermoscientific)) inserted in the BamHI/XbaI sites of pRK5-Myc (Clontech). The small 3′-UTR fragment was included to facilitate the PCR primer design and it does not contain a miR-517a target site.

## Quantitative PCR

Expression level of endogenous miRNA or those reversely transfected as described above was analyzed as follows. Total RNA was isolated with the TRIzol reagent according to the manufacturer's instructions (Life Technologies, Cat# 15596018) and transcribed into complementary DNA (cDNA) with TaqMan microRNA Reverse Transcription kit (Life Technologies, Cat# 4366596). Gene expression was quantified by TaqMan Gene Expression Master mix (Life Technologies, Cat# 4369016) on a LightCycler 4800 RT–PCR System (Roche Applied Science, Germany). Relative amounts of miRNA between samples were calculated with the comparative CT method with normalization to the RNU6B control CT value. TaqMan probes used: hsa-miR-146a-5p (Life Technologies, Assay ID 000468, Cat# 4427975); has-miR-505-3p (Life Technologies, Assay ID 002089, Cat# 4427975); and RNU6B (Life Technologies, Assay ID 001093, Cat# 4427975).

## Data availability

### Primary data

In addition to the provided dataset files, source data in this study have been made available at:
Shields *et al*, microRNA Induction of Fatal Differentiation Programs is a Context-selective Vulnerability in Ovarian Cancer Cells. Gene

Expression Omnibus (GEO). Accession Number: GSE67330.

Exome sequencing data are available at the NCBI SRA Accession number: SRP065357.

CCLE data: http://www.broadinstitute.org/ccle

Barretina J *et al*, The Cancer Cell Line Encyclopedia enables predictive modeling of anticancer drug sensitivity. *Nature* 2012 Mar 29; 483(7391): 603–607.

TCGA Data: http://tcga-data.nci.nih.gov/

Cancer Genome Atlas Research N. Integrated genomic analyses of ovarian carcinoma. *Nature* 2011 Jun 30; 474(7353): 609–615.

## Cited data

Byers LA *et al*, An epithelial-mesenchymal transition gene signature predicts resistance to EGFR and PI3K inhibitors and identifies Axl as a therapeutic target for overcoming EGFR inhibitor resistance. *Clin Cancer Res* 2013 Jan 1; 19(1): 279–290.

Garnett MJ, Edelman EJ, Heidorn SJ, Greenman CD, Dastur A, Lau KW, *et al* Systematic identification of genomic markers of drug sensitivity in cancer cells. *Nature* 2012 Mar 29; 483(7391): 570–575.

**Expanded View** for this article is available online.

## Acknowledgements

This study was supported by grants from the NIH (CA71443, CA129451, P50CA083639, U54CA151668, T32-GM008203), The Robert Welch Foundation (I-1414), and CPRIT (RP110595, RP110763). The authors would like to thank Drs. Hani Gabra, William Hahn, Robert Bast, and John Abrams for their generous donation of cell lines. The data discussed in this publication have been deposited in NCBI's Gene Expression Omnibus (Edgar *et al*, 2002) and are accessible through GEO Series accession number GSE67330 (http://www.ncbi.nlm.nih.gov/geo/query/ acc.cgi?acc=GSE67330).

## Author contributions

BBS was the primary author, designed and performed multiple experiments, and interpreted the data; CVP performed mouse xenograft experiments, HG designed and performed multiple experiments including qPCR; EM performed data analytics, MP provided source data for miRNA screens in NSCLC line; CN performed miRNA mimic screens of newly derived polyclonal cell lines; SP performed miRNA mimic screens of PEO1 and PEO4 lines; YW and CI performed TCGA data processing and analysis; HSK performed data processing for next generation sequencing; RJB generated the CIRCOS plot; SK performed whole-exome sequencing and RNA sequencing of PEO1 and PEO4 lines; CRA and GLB assisted with DOPC particle production; JL and AG generated newly derived polyclonal cell lines, KAB supervised TCGA data processing; AKS was responsible for experimental design of mouse xenograft experiments and was a chief collaborator; MAW was the corresponding author and was responsible for experimental design and data interpretation.

## Conflict of interest

The authors declare that they have no conflict of interest.

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
