## [Review Process File · Molecular Systems Biology]

A genome-scale screen reveals context-dependent ovarian cancer sensitivity to miRNA overexpression

Benjamin B. Shields, Chad V. Pecot, Hua Gao, Elizabeth McMillan, Malia Potts, Christa Nagel, Scott Purinton, Ying Wang, Cristina Ivan, Hyun Seok Kim, Robert J. Borkowski, Shaheen Khan, Cristian Rodriguez-Aguayo, Gabriel Lopez-Berestein, Jayanthi Lea, Adi Gazdar, Keith A. Baggerly, Anil K. Sood and Michael A. White

Corresponding author: Michael A. White, UT Southwestern Medical Center

Review timeline:

Submission date:	18 May 2015
Editorial Decision:	26 June 2015
Revision received:	23 September 2015
Editorial Decision:	21 October 2015
Revision received:	05 November 2015
Accepted:	09 November 2015

Editor: Maria Polychronidou

Transaction Report:

1st Editorial Decision

26 June 2015

Thank you again for submitting your work to Molecular Systems Biology. We have now heard back from the two referees who agreed to evaluate your manuscript. As you will see from the reports below, the referees acknowledge that the study seems potentially interesting. However, they raise a series of concerns, which should be carefully addressed in a revision of the manuscript. The referees' recommendations are rather clear so there is no need to repeat the points listed below.

If you feel you can satisfactorily deal with these points and those listed by the referees, you may wish to submit a revised version of your manuscript. Please attach a covering letter giving details of the way in which you have handled each of the points raised by the referees. A revised manuscript will be once again subject to review and you probably understand that we can give you no guarantee at this stage that the eventual outcome will be favorable.

REFeree COMMENTS

Reviewer #1:

The manuscript by Shields et al. studied the cell proliferation/survival effects of overexpression of miRNAs (by synthetic miRNA mimetics) on a panel of 16 ovarian cancer cell lines and two additional control cell types. Through this extensive effort and more detailed functional and genomic characterization, the authors found that most miRNAs affect a small number of cell lines,

with maybe the exception of miR-517a and miR124. This supports functional diversity of ovarian cancer cell lines and their molecular vulnerability. The authors further characterized the molecular mechanisms of these miRNAs.

Overall, the authors produced a substantial amount of data. Although the conclusions may not necessarily be the most exciting, the test of cellular susceptibility to a large number of miRNA mimetics across a large number of cell types is currently missing in the field, and thus this study is of value.

The study was overall carefully performed. A few relatively minor issues are noted below.

1. The screen procedure involves selecting with puromycin from 2 days after mimetics transfection. It is not very obvious why there needs to be a puromycin selection process, as Dharmacon miRNA mimetics do not have puromycin resistance.
2. The conclusion that functional diversity of ovarian cancer cell lines underlie the variable responses toward miRNA mimetics in different cell lines is heavily dependent on data in Fig 1A. Fig 1A in turn depends on the assay variability. Although the authors showed variation of control assays by multiple replicates, it will be useful to know the variability of the viability assay data for real miRNAs. I suggest the authors to take 5-10 miRNAs to re-examine across 5-10 cell lines.
3. As miRNA mimetics tend to drive superphysiological levels of ectopic miRNA expression, it is not very easy to interpret how much miRNAs were overexpressed by fold changes (Fig S1C, S1D). I suggest the authors to quantify the absolute molecules of miRNAs per cell as a read out for those two figures. This can be achieved by using synthetic miRNAs as standards in quantification.

Reviewer #2:

This manuscript describes the identification of multiple miRNAs that upon (over) expression induce lethality in patient derived ovarian cancer cell lines. The results indicate a high degree of heterogeneity across a panel of cell lines with respect to the response to the expression of specific miRNAs. For the SKOV3 cell line specific toxic miRNAs (miR-146a and miR-505) the authors observe a correlation between increased miR-146a and miR-505 expression and increased overall survival in TCGA datasets. In addition the authors identify miR155 and miR181b, selectively reducing the viability of chemo-resistant PEO4 cells and not of several other cell lines. They implicated both miRNAs in the control of AKT pathway although the effects seem not to completely recapitulated with AKT inhibitor. Despite the cell line specificity they extend the effects of miR155 and miR181b into panels of lung cancer cells lines and breast cancer cell lines, showing a correlation between miR155 and miR181b expression and sensitivity to the PI3K pathway inhibitor GDC0941. They also implicate miR155 and miR181b in TGF β signaling and EMT and show sensitivity to combinatorial treatment with both TGF β and AKT inhibitors. The observation of the correlation between EMT and sensitivity is extended in a panel lung cancer cell lines where a correlation exists between miR155 and miR181b sensitivity and the expression epithelial markers. Finally, the authors focus their analyses on miR-517a and miR-124. Through expression profiling and target predictions they implicate ARCN1 and SIX4 as targets for respectively miR-517a and miR-124. Further analysis suggest potential pathways and multiple biological pathways affected by these genes potential contributing to the observed phenotypes. Finally, they use xenograft models in combination with DOPC neutral liposomes with siRNA to show reduction in tumor growth for miR-517a and siRNA SIX4. A final conclusion of this work is that the re-activation of specific miRNAs can induce cell differentiation programs as a shared mechanism for the toxic effect of several of the miRNAs studied in this work.

The authors are to be complimented with the amount of work, the extensive integration of multiple large-scale datasets and the exploration of several different biological mechanisms in this study. However, as a consequence it is difficult to judge the ways by which the different miRNAs were selected for follow-up. One could argue that this has been a reversed selective process and only those examples that did show a significant correlation in other datasets are presented here. For example, one selection has been the miRNAs selective for SKOV3. How about the top hits in other cell lines in this panel? Did they have the same type of correlations in other (clinical) datasets? Also the correlation between the effect of miRNA sensitivity and siRNA toxicity seems a fruitful approach (Figure 3D) but raises the issue of multiple testing corrections. How many of these

examples were identified across the whole panel and how many were significant (before and after multiple testing correction)? The authors mention that the effect of USP1 independent is of miR-517a. Does this finding also invalidate the approach taken for the identification of ARCNI (Figure S5A versus 3F)?

With respect to the xenograft experiments, one would like to see the inclusion of cell lines that are insensitive to the effects of either the miR-517a or siSIX4. These cell lines are present in the panel and should be used as negative controls for the experimental set-up in vivo.

Another concern is the pleiotropic effects of miRNAs. The authors point this out "These observations indicate that supra physiological concentrations of miRNAs have highly pleiotropic consequences on cellular gene expression programs, and therefore likely influence biological processes via highly combinatorial mechanisms". However, they do attempt to pinpoint the effects to individual genes and consequently state that "only SIX4 was sufficient to mimic the consequences of miR-124 in ovarian cancer cells". The authors should address this apparent discrepancy.

The authors state that the heterogeneity with respect to the effects of the different miRNAs is due to the molecular etiology of ovarian cancer. One could argue that this could be demonstrated by the clustering of the cell line panel according to molecular subtypes or other characteristics. The authors have not addressed this possibility of a different type of classification. It is a particular interest for this work to take the expression patterns of all miRNAs in the cell lines in this panel and try to identify signatures or subgroups (differentiation grades) related to the sensitivity towards specific miRNAs targeting pathways connected to subtypes.

Minor points:

Many of the findings are based on a small set of cell lines (PEO1 and PEO4 and 3 polyclonal cell populations). Is there the possibility to extend these numbers to make it less specific and prone to context dependency?

It is unclear what is presented in Figure 2A: is this miR-155? miR181b or both?

The authors suggest "differentiation therapy" as an option for treating ovarian cancer. Although appealing one could ask what is really meant by this, how do they deal with the (also in this paper described) heterogeneity and subtypes. The example of RA induced differentiation does not really relate to solid cancers including ovarian cancer.

1st Revision - authors' response

23 September 2015

Reviewer 1

The study was overall carefully performed. A few relatively minor issues are noted below.

1. The screen procedure involves selecting with puromycin from 2 days after mimetics transfection. It is not very obvious why there needs to be a puro selection process, as Dharmacon miRNA mimetics do not have puro resistance.

Response: There was no puro selection process during the screen. Puro selection was only used for complementation assays with cDNA expression vectors containing miRNA targets and the puro-resistance marker. We have separated this section of the methods into an independent paragraph in the revised manuscript in order to help make this clear.

2. The conclusion that functional diversity of ovarian cancer cell lines underlie the variable responses toward miRNA mimetics in different cell lines is heavily dependent on data in Fig 1A. Fig 1A in turn depends on the assay variability. Although the authors showed variation of control assays by multiple replicates, it will be useful to know the variability of the viability assay data for real miRNAs. I suggest the authors to take 5-10 miRNAs to re-examine across 5-10 cell lines.

Response: We did not effectively communicate the work we have done assessing assay variability, and we greatly appreciate the reviewer catching this. The primary screens were performed as biological triplicates. All miRNAs were tested 3X in all cell lines. In the revised manuscript, we provide an extended data set reporting all values for all replicates together with the means and standard deviations (new Table 1). In addition, we have included a new supplemental figure (now supplemental figure 1B) that displays the standard deviation distributions among biological triplicates across the cell line panel (black curve). In addition, we display the phenotypic variation among miRNA seed family members (red curve) relative to the total phenotypic variation (blue curve). These distributions indicate high reproducibility among biological triplicates (black versus blue) and high phenotypic correlation among seed family miRNA mimics (red versus blue):

Legend: For each microRNA mimic, a standard deviation value was calculated for viability across the panel of cell lines (blue curve). For all miRNA's in the same seed family, a standard deviation value was calculated for viability across the panel of 16 cell lines (red curve). Lastly, a within-replicate standard deviation value was calculated (black curve). A kernel density estimation was fit to each of the three standard deviation distributions and plotted.

3. As miRNA mimetics tend to drive superphysiological levels of ectopic miRNA expression, it is not very easy to interpret how

much miRNAs were overexpressed by fold changes (Fig S1C, S1D). I suggest the authors to quantify the absolute molecules of miRNAs per cell as a read out for those two figures. This can be achieved by using synthetic miRNAs as standards in quantification.

Response: We completely agree that mimetics drive supraphysiological levels of ectopic miRNA expression. We also agree that the data display for the original Fig. S1C, S1D was suboptimal for making that point. Therefore, in the revised manuscript, we have replaced these figures with bar graphs indicating endogenous miRNA abundance in each cell line and ectopic miRNA abundance in each cell line (rather than fold changes). Both of these (now Figures S2B and S2C) were derived by the comparative CT method using RNU6B for normalization.

Reviewer 2

The authors are to be complimented with the amount of work, the extensive integration of multiple large-scale datasets and the exploration of several different biological mechanisms in this study. However, as a consequence it is difficult to judge the ways by which the different miRNAs were selected for follow-up. One could argue that this has been a reversed selective process and only those examples that did show a significant correlation in other datasets are presented here. For example, one selection has been the miRNAs selective for SKOV3. How about the top hits in other cell lines in this panel? Did they have the same type of correlations in other (clinical) datasets? Also the correlation between the effect of miRNA sensitivity and siRNA toxicity seems a fruitful approach (Figure 3D) but raises the issue of multiple testing corrections. How many of these examples were identified across the whole panel and how many were significant (before and after multiple testing correction)? The authors mention that the effect of USP1 independent is of miR-517a. Does this finding also invalidate the approach taken for the identification of ARCNI (Figure S5A versus 3F)?

Response: We have tried to be explicit about the rationale for follow-up: selective activity in the patient-matched chemo-resistant model (miRs 155 and 181); common activity in the traditional cell models (miR-517); and uniform activity in the heterogeneous “explant” cultures (miR-124). This was based on somewhat arbitrary biological interest given the observed activity distributions. With respect SKOV3-toxic miRs – only one clinical data set was available with reliable patient outcome data. Therefore Keith Baggerly split the cohort into a 2/3 training set and 1/3 validation set prior to

any correlation analysis. With respect to the miRNA/siRNA sensitivity correlations—we completely agree that this approach is susceptible to false discovery due to multiplicity of testing. It is important to note that this is a heuristic, not a stand-alone finding, to which we applied biological testing in order to establish relevance. In this particular case, the multiplicity of testing was also low. The methods section of the revised manuscript has been revised to include a more detailed description of this analysis: “Identification of potentially active miR-517a targets was achieved as follows. A total of 52 miR-517a target genes were predicted by TargetScan 6.0 (www.targetscan.org). These predicted targets were then filtered for activity (relative cell viability < 0.5) H2122 with known sensitivity to miR-517a that had also previously undergone a genome-wide siRNA-toxicity screen. Toxicity in response to depletion of each of these 10 predicted targets, with two independent siRNA pools, was then assessed for correlation with toxicity upon transfection with miR-517a mimic in a panel of 13 NSCLC lines. Only 2 predicted targets, USP1 and ARCNI, demonstrated a positive correlation with miR-517a ($R^2 > 0.35$) using siRNA oligos from both Dharmacon and Ambion”. With respect to USP1- our observations do not indicate the effect of USP1 is independent of miR-517a—in fact we show mechanistic consequences on ID1. The difference is that the relationship is much more context-selective than ARCNI in ovarian cancer cells, and may be more commonly encountered in lung cancer cells. S5A shows lung cancer cell lines, and 3F shows ovarian cancer cell lines.

The authors state that the heterogeneity with respect to the effects of the different miRNAs is due to the molecular etiology of ovarian cancer. One could argue that this could be demonstrated by the clustering of the cell line panel according to molecular subtypes or other characteristics. The authors have not addressed this possibility of a different type of classification. It is a particular interest for this work to take the expression patterns of all miRNAs in the cell lines in this panel and try to identify signatures or subgroups (differentiation grades) related to the sensitivity towards specific miRNAs targeting pathways connected to subtypes.

Response: This is a good idea and do-able using mRNA expression profiles—and has been included in the revised manuscript as Supplemental figures 1C, 1D, 1E, and 1F. In brief, we used affinity propagation clustering to identify deterministic patterns of similarity among the miRNA mimic phenotypes across the cell line panel (new figure S1C); and among the cell lines based on whole genome transcript profiles (RNAseq, new figure S1E), or miRNA mimic sensitivity profiles (new figure S1D). This revealed at least 50 phenotypic miRNA clusters that corresponded to 5 distinct cell line clusters. At least 4 mRNA expression-based subtypes are present within the cell panel. However, these clusters had unimpressive correspondence to miRNA viability phenotype-based clusters (new Figure S1F) indicating global gene expression phenotypes, considered as a whole, did not specify selective response to the miRNA mimic library.

With respect to the xenograft experiments, one would like to see the inclusion of cell lines that are insensitive to the effects of either the miR-517a or siSIX4. These cell lines are present in the panel and should be used as negative controls for the experimental set-up in vivo.

Response: The major intent of the xenograft assays was to examine if the toxicity phenotype was synthetic to plastic or not. The control here was an innocuous miRNA mimic in the same model as a toxic miRNA mimic. We have described these observations with care to avoid over interpretation. The experiment suggested by the referee could be informative, but is really asking a separate question, and would require extensive resource and time-intensive follow-up to be meaningful.

Another concern is the pleiotropic effects of miRNAs. The authors point this out "These observations indicate that supra physiological concentrations of miRNAs have highly pleiotropic consequences on cellular gene expression programs, and therefore likely influence biological processes via highly combinatorial mechanisms". However, they do attempt to pinpoint the effects to individual genes and consequently state that "only SIX4 was sufficient to mimic the consequences of miR-124 in ovarian cancer cells". The authors should address this apparent discrepancy.

Response: We did not intend to suggest that “only SIX4 depletion was sufficient to mimic the consequences of the miR-124 in ovarian cancer cells”, and I have made certain that statement does not appear anywhere in the manuscript. We were only describing the experimental observation that SIX4 depletion was sufficient to mimic the selective toxicity.

Minor points:

Many of the findings are based on a small set of cell lines (PEO1 and PEO4 and 3 polyclonal cell populations). Is there the possibility to extend these numbers to make it less specific and prone to context dependency?

It is unclear what is presented in Figure 2A: is this miR-155? miR181b or both?

The authors suggest "differentiation therapy" as option for treating ovarian cancer. Although appealing one could ask what is really meant by this, how do they deal with the (also in this paper described) heterogeneity and subtypes. The example of RA induced differentiation does not really relate to solid cancers including ovarian cancer.

Response: 1. PEO1/PEO4 are the only patient-matched chemo-sensitive/chemo-resistant model we have available. For that reason, we confined those particular studies to the biological impact of the miRNA mimics (further examined in 41 lung cancer lines and 6 breast cancer lines) and were careful not to make any generalizable statements about mechanisms of platinum resistance in ovarian cancer. 2. Both miR-155 and miR-181b are presented in Figure 2A. We have re-plotted the source data using two different colors in the revised manuscript (new figure 2A) to illustrate this more effectively. 3. We appreciate the referee's caution, and have modified the corresponding text in the revised discussion section accordingly: "...Thus, the common loss of miRNA expression and maturation in ovarian cancer cells might serve to deflect anomalous engagement of cellular differentiation programs in response to oncogene activation; offering differentiation therapy for consideration as a potential treatment modality for ovarian cancer. Though distinct from the solid tumor context, perhaps the most well-known example of differentiation therapy is the treatment of acute promyelocytic leukemias (APMLs) with all trans-retinoic acid (ATRA)..."

2nd Editorial Decision

21 October 2015

Thank you again for submitting your work to Molecular Systems Biology. We have now heard back from the referee who agreed to evaluate your manuscript. As you will see below, this referee is now satisfied with most of the modifications made, but s/he is still concerned about the lack of xenograft experiments involving a cell line that is non-responsive to mir-517a. We think that including such experiments is not mandatory for the acceptance of this work, since they would not provide sufficient insights into the subtype specificity of mir-517a, unless several cell lines representative of different subtypes would be analyzed. However, it should be clearly mentioned in the text (as the reviewer recommends) that as it stands these xenograft experiments simply "validate the effect of mir-517a in SKOV3 cells".

Before formally accepting the manuscript, we would like to ask you to address some editorial issues listed below.

 REFEREE COMMENTS

Reviewer #2:

The authors have addressed the comments of the reviewers in an adequate manner. They are to be complimented for the additional analyses performed although they have unexpectedly not yielded further insight in the observed cell line dependencies.

One issue remains with the extension of the effect of miR-517a to the in vivo model. I do not share the reasoning that the xenograft experiments do not require a non-responsive control. Although I agree that the xenograft experiment is designed to recapitulate the phenotype observed in vitro, it does suggest that the EOC specific miR-517a or SIX4 phenotypes do extend to the in vivo

setting. The authors describe this experiment as "To model miR-517a sensitivity within tumors" but I would argue that it should be rephrased to "to validate the effect of mir-517a in SKOV3 cells, we use a mouse xenograft model". This point that miR-517a has a subtype-specific effect has not been formally proven by the experiments presented in this manuscript.

2nd Revision - authors' response

05 November 2015

Response to Reviewer #2:

The authors have addressed the comments of the reviewers in an adequate manner. They are to be complimented for the additional analyses performed although they have unexpectedly not yielded further insight in the observed cell line dependencies.

One issue remains with the extension of the effect of miR-517a to the in vivo model. I do not share the reasoning that the xenograft experiments do not require a non-responsive control. Although I agree that the xenograft experiment is designed to recapitulate the phenotype observed in vitro, it does suggest that the EOC specific miR-517a or SIX4 phenotypes do extend to the in vivo setting. The authors describe this experiment as "To model miR-517a sensitivity within tumors" but I would argue that it should be rephrased to "to validate the effect of mir-517a in SKOV3 cells, we use a mouse xenograft model". This point that miR-517a has a subtype-specific effect has not been formally proven by the experiments presented in this manuscript.

This reviewer's point is well taken and the text of the manuscript has been updated to directly state that the xenograft experiments only validate the effect of miR-517a in SKOV3 cells.